# Unsupervised Learning via Network-Aware Embeddings

## Abstract

Data clustering, the task of grouping observations according to their similarity, is a key component of unsupervised learning – with real world applications in diverse fields such as biology, medicine, and social science. Often in these fields the data comes with complex interdependencies between the dimensions of analysis, for instance the various characteristics and opinions people can have live on a complex social network. Current clustering methods are ill-suited to tackle this complexity: deep learning can approximate these dependencies, but not take their explicit map as the input of the analysis. In this paper, we aim at fixing this blind spot in the unsupervised learning literature. We can create network-aware embeddings by estimating the network distance between numeric node attributes via the generalized Euclidean distance. Differently from all methods in the literature that we know of, we do not cluster the nodes of the network, but rather its node attributes. In our experiments we show that having these network embeddings is always beneficial for the learning task; that our method scales to large networks; and that we can actually provide actionable insights in applications in a variety of fields such as marketing, economics, and political science. Our method is fully open source and data and code are available to reproduce all results in the paper.

## 1 Introduction

Finding patterns in unlabeled data – a task known as unsupervised learning – is useful when we need to build understanding from data Hastie et al. (2009). Unsupervised learning includes grouping observations into clusters according to some criterion represented by a quality or loss function Gan et al. (2020) – data clustering. Applications range from grouping of genes with related expression patterns in biology Ranade et al. (2001), finding patterns in tissue images in medicine Filipovych et al. (2011), or segment customers for marketing purposes.

Popular data clustering algorithms include DBSCAN Ester et al. (1996), OPTICS Ankerst et al. (1999), k-Means, and more. Modern data clustering approaches rely on deep learning and specifically deep neural networks Aljalbout et al. (2018); Aggarwal et al. (2018); Pang et al. (2021); Ezugwu et al. (2022), or denoising with autoencoders Nawaz et al. (2022); Cai et al. (2022). However, these approaches work in (deformations of) Euclidean spaces – where dependencies between the dimensions of the analysis can be learned Mahalanobis (1936); Xie et al. (2016) –, but the problem to be tackled here is fundamentally non-Euclidean Bronstein et al. (2017). Graph Neural Networks (GNN) Scarselli et al. (2008); Wu et al. (2022); Zhou et al. (2020a) work in non-Euclidean settings, and they are the focus of this paper.

To see why, consider product adoption in a social network – with an example in Figure 1. We want to find product clusters depending on the people who buy them. However, the purchase decision of each person is influenced by their acquaintances in a complex social network. By using the information in the social network, we could cluster what could have appeared as otherwise independent vectors. In Figure 1 products (a) and (b) are clearly related to each other and so are products (c) and (d).

To perform this clustering task we need to generate network-aware embeddings: to use the network's topology as the space in which observations live, which is the basis to estimate their similarities and, ultimately, their clusters. This is the main objective of this paper: to cluster node attributes on a complex network. We base our solution on previous research that established ways to estimate the

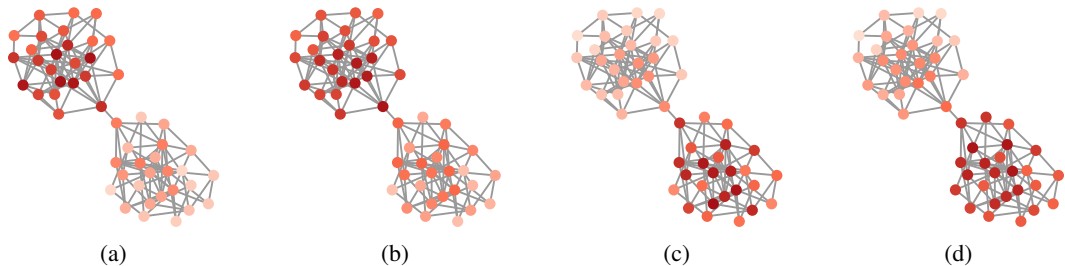

| (a) | (b) | (c) | (d) |

Figure 1: A toy example of product adoption in a social network. Nodes are people, connected to their friends. Node color determines how strongly they adopt a product (dark = high engagement, light = low engagement). (a-d) Different products.

distance Coscia et al. (2020); Coscia (2020; 2022) and (co)variance (correlation) Coscia (2021b); Devriendt et al. (2022) between numeric node attributes on a complex network.

The contributions of this paper are threefold. First, our problem definition is innovative. GNNs almost universally share the assumption that the entities worth analyzing are the nodes of the network, and that their attributes refer to the same entity as the node. This is not the case here: node attributes are entities in their own right, and the nodes of the graph represent the *dimension* of the analysis, not the observations. As a result, when used in tasks related to clustering, GNNs are mostly used to find clusters of nodes Bo et al. (2020); Tsitsulin et al. (2020); Bianchi et al. (2020); Zhou et al. (2020b). GNN-based clustering seeks to find node embeddings Perozzi et al. (2014b); Hamilton et al. (2017), but we are interested in finding node *attribute* embeddings. When node attributes are taken into account in GNNs, they always serve the purpose of aiding the classification of nodes rather than clustering the attributes themselves Perozzi et al. (2014a); Zhang et al. (2019); Wang et al. (2019); Lin et al. (2021); Cheng et al. (2021); Yang et al. (2023), which is not the objective here.

GNN clustering is an evolution of the classical problem of community discovery Fortunato (2010); Rossetti & Cazabet (2018); Fortunato & Hric (2016). To the best of our knowledge, there are no known cases of algorithms dedicated to cluster observations whose dimensions can be mapped on a complex network structure by using that network structure to generate embeddings. The community discovery literature shares with GNN clustering the use of node attributes to classify the nodes Leskovec et al. (2010); Yang et al. (2013); Bothorel et al. (2015); Chunaev (2020) or provide a ground truth for the communities Peel et al. (2017), not to cluster the attributes themselves.

Second, we create a pipeline integrating a distance measure between observations on a graph with a full data clustering process. To the best of our knowledge, this is the first pipeline directly addressing the problem we want to study: to cluster node attributes.

Finally, we show in our experimental section that our node attribute clustering pipeline performs better than the alternatives on synthetic data and real world data with a ground truth. Having network embeddings is always beneficial for the learning task and can enhance dimensionality reduction techniques such as t-distributed Stochastic Neighbor Embedding (tSNE) by providing a more accurate depiction of the complex space in which the observations live. Our network embeddings can also improve deep learning techniques such as graph autoencoders. We also show that calculating network embeddings with our technique is scalable and we present a few case studies showing how our method can be applied in such diverse fields as macroeconomics, politics, and marketing.

The code and data to reproduce our results is available as supplementary material and on the web[1].

## 2 METHODOLOGY

### 2.1 DATA MODEL

The framework, illustrated in Figure 2, needs two main components: a graph $G$ and the set of observations $O$ we want to classify into clusters.

---

[1][URL redacted for double blind review]

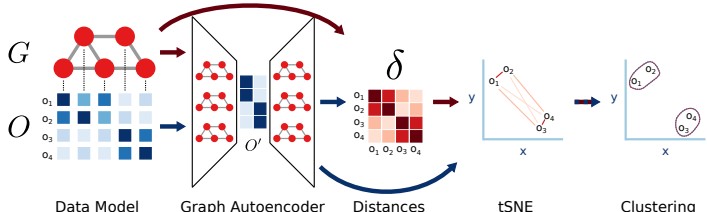

Figure 2: Our full workflow. Red colors track the flow of the information coming from the graph, and blue colors track the information coming from the observations.

The graph $G = (V, E)$ is composed by a set of nodes $V$ and a set of edges $E \subseteq V \times V$. Each edge is a pair of nodes $(u, v) \in V$. The edges can be weighted, i.e. they can be triples $(u, v, w)$, with $w \in R^+$ being any non-negative weight. The weight represents the capacity of an edge Coscia (2021a), meaning that the higher the weight the closer the two connected $u, v$ nodes are. We can build network embeddings on multi-layer networks Coscia (2022), networks with multiple different qualitative types of edges. This means that an edge could be represented by a quadruple $(u, v, w, t)$, with $t \in T$ representing the type (layer) of the edges.

One necessary requirement is that $G$ must have a single connected component: all pair of nodes in $G$ needs to be reachable via paths through the edges of $G$. The embeddings cannot be calculated in networks with disconnected components. We also need the absence of self-loops, i.e. edges connecting a node to itself. For simplicity, we work with undirected graphs, i.e. $(u, v) = (v, u)$.

As for $O$ this is a set of observations or data points. Each observation $o \in O$ is a vector of length $|V|$ – i.e. it is a node attribute assigning a value for each node $v \in V$. One can consider $V$ as being the dimensions of each observation in $O$ and $G$ being an object that describes the complex interdependencies between these dimensions.

## 2.2 Problem Definition

We now formally define the problem we are intending to solve, as it is different from the classical approach of graph neural networks and graph clustering.

**Definition 1** (Problem Definition). Let $G = (V, E)$ be a connected undirected graph, with $V$ being the nodeset and $E \subseteq V \times V$. Let $O$ be a set of numerical vectors of length $|V|$ – the attributes of the nodes of $G$. Find the function $f : G \times O \to P$ returning the partition $P$ such that $\arg\min_O \delta = f(G, O)$, $\delta$ being the function calculating the distance between pairs of observations on $G$.

In other words, we want to find the partition $P$ of $O$ such that the graph distance $\delta$ over $G$ of observations within the same group in $P$ is minimized – excluding trivial solutions that put each observation in a singleton cluster. This definition hinges on $\delta$: the ability to calculate the graph distances between two $o_1, o_2 \in O$. There are many possible non-trivial options for $\delta$, and the next section provides a reasonable one as one of the main contributions of this paper.

## 2.3 Network Distances

One key step to perform unsupervised learning via clustering is to estimate the distances between the observations. That is, given observations $o_1$ and $o_2$, we want to have a function $\delta_{o_1,o_2}$ quantifying the distance between them. Sufficiently close observations may be part of the same cluster.

One could get better results by transforming observations in $O$ so that their noisy estimates can be better handled by $\delta$ – see Section 2.4. Here instead we consider the fact that one could choose a different $\delta$ function that better conforms to one's expectation of proximity between observations. The simplest case is using the Euclidean distance, which assumes that all dimensions used to record observations in $O$ are independent and equally important. Here, we assume that observations in $O$ live in a complex space with interdependencies between the dimensions of analysis mapped by a graph $G$. If this assumption is correct, then the distance between $o_1$ and $o_2$ needs to take $G$ into account: to estimate how far two observations are we need to know how to traverse $G$ to move

from the $o_1$ position to the $o_2$ position in this complex space. We want to calculate a Generalized Euclidean (GE) distance, that can take any possible dimension interdependency into account.

Notation-wise, our functions becomes $\delta_{o_1,o_2,G}$, since it requires $G$ to be estimated. For this paper, $\delta_{o_1,o_2,G}$ is based on a solution Coscia (2020) to the node vector distance problem Coscia et al. (2020). In GE, one can use the pseudoinverse Laplacian ($L^+$) to calculate the effective resistance between two arbitrary $o_1$ and $o_2$ vectors. The Laplacian matrix is $L = D - A$, with $A$ being the adjacency matrix of $G$ and $D$ being the diagonal matrix containing the degrees of the nodes of $G$:

$$\delta_{o_1,o_2,G} = \sqrt{(o_1 - o_2)^T L^+ (o_1 - o_2)}.$$

Previous work shows that this formula gives a good notion of distance between $o_1$ and $o_2$ on a network Coscia (2020). For instance, it can recover the infection and healing parameters in a Susceptible-Infected-Recovered (SIR) model by comparing two temporal snapshots of an epidemic.

Calculating $L^+$ is computationally expensive, in the order of $\mathcal{O}(|V|^3)$ but we do not need to compute it explicitly, as we show in Section 3.4. We can also work with multilayer networks – networks with multiple qualitatively different types of edges Kivelä et al. (2014); Boccaletti et al. (2014) – by defining a multilayer $L$. This is achieved by calculating the Laplacian of the supra-adjacency matrix Porter (2018); Coscia (2022). We can define $B$ as a $|V| \times |E|$ incidence matrix telling us which node is connected to which edge. Then, $L = BWB^T$, with $W$ being the diagonal matrix containing the weights of each edge $e \in E$. In this case, $E$ can contain both regular intra-layer edges as well as the inter-layer couplings connecting nodes from one layer to nodes in the other layers.

## 2.4  CLUSTERING

Following Figure 2, we now describe all remaining components of the framework. Note that, with the exception of the clustering step, none of the components is strictly speaking mandatory: each can be removed and we can still cluster the data. However, each step performs a useful function and has a role in improving the final result – as Section 3.2 shows.

The logical steps are: clean noise and then reduce the dimensions in $O$ to get better-separated clusters that are easier to find with a classical clustering algorithm. Both steps should use information from $G$. For this reason, the first step (cleaning noise) is done via a Graph Autoencoder (GAE) Kramer (1991); Kipf & Welling (2016b); and the second step (dimensionality reduction) via tSNE Van der Maaten & Hinton (2008) using GE as the spatial metric instead of a non-network metric.

### 2.4.1  CLEANING NOISE

An autoencoder (AE) creates embeddings generated with a deep neural network formed by an encoder and a decoder. Since for the hidden layers we use graph convolution Kipf & Welling (2016a) on $G$, the AE is actually a GAE. Our choice of graph convolution for the hidden layers – both encoder and decoder – is the GraphSAGE Hamilton et al. (2017) operator, with SoftSign activation function and a sigmoid normalization of the last layer of the decoder. The autoencoder is trained via backpropagation using cross entropy loss. We could use different activation functions, and different graph convolution approaches – for instance Graph Attention Networks Velickovic et al. (2017). We picked our components as they are the ones performing the best in our validation.

### 2.4.2  DIMENSIONALITY REDUCTION

tSNE creates shallow embeddings and works best when reducing to a low number of dimensions – here we set it to two. We could apply tSNE directly to the GAE output. However that would mean that tSNE is seeking for the best representation of the data in an Euclidean space, which is not appropriate because we know the dimensions of our observations are related to each other in $G$. Luckily, the tSNE algorithm is agnostic to the function used to calculate the distance between two observations. We can provide our GE function as the metric over which tSNE operates.

The role of GE is to take the complex interdependencies between dimensions expressed by the graph $G$ into account for tSNE. In this way, tSNE can optimize cluster separation in the complex space

defined by $G$. If the real clusters of $O$ are correlated with $G$'s topology, using GE instead of any other metric space will lead to a significant performance increase.

### 2.4.3 CLUSTER DETECTION

The last step is to perform the clustering itself. We choose DBSCAN Ester et al. (1996) due to its simplicity, low time-complexity, and ability to find non-convex clusters of arbitrary shapes.

When refer to the full framework as GAE+GE+tSNE. If we do not perform the noise cleaning step, then our framework becomes GE+tSNE. We can also use an Euclidean space to perform tSNE, skipping the GE step and obtaining GAE+tSNE.

DBSCAN also needs to define in which metric space to operate — just like tSNE. So one could use the GE metric space here as well. However, this cannot be done if we performed dimensionality reduction with tSNE, because GE can only work on the original dimensions in $G$. For this reason, a GAE+GE+tSNE+GE is impossible. One could make a GAE+GE framework, skipping tSNE and using GE directly in DBSCAN. However, in Section3.2 we show that the synergy between GE and tSNE is strong and it is the factor that drives the performance.

Some of the components of our framework could be swapped with others, to maximize the performance in specific settings. For instance, one could replace the GAE to clean noise with a generative adversarial network Creswell et al. (2018), or tSNE to reduce dimensionality with principal component analysis (PCA) or non-negative matrix factorization, or the DBSCAN clustering step with OPTICS, k-Means or any other specialized clustering technique. However, this is not a relevant dimension of analysis for this paper, because none of these alternative components could replace the network embeddings we provide with GE, which is the fundamental contribution of this paper, and this is why we do not test how much, e.g., using PCA instead of tSNE can improve the performance.

## 3 EXPERIMENTS

### 3.1 SETUP

To contextualize the performance of our framework in a network-aware clustering, we perform our validation tests in two-steps. First, we investigate the performance of each method in isolation. The methods we consider are either the various parts of our framework, or potential alternative methods. To sum them up, the isolated components/baselines are (clusters are always extracted via DBSCAN):

- Baseline: clusters $O$ using an Euclidean space (no network information, no $O$ preprocess).
- GE: clusters $O$ using the GE space.
- tSNE: clusters $O$ by first reducing each observation to two dimensions using tSNE.
- N2V: clusters $O$ multiplied by the node embeddings obtained from node2vec Grover & Leskovec (2016) (we set $p = q = 1$, making this equivalent to DeepWalk Perozzi et al. (2014b), since different $p$ and $q$ values did not lead to significantly different results).
- GAE: clusters $O$, after passing it through our graph autoencoder, described in Section 2.4.

Since these methods can be combined in a larger framework, as we do in Section 2.4, in the second step we do so. In practice, the second step is an ablation study where we investigate the effect of removing each component from the framework. Since GAE is the method performing the best in isolation, it is taken as the baseline for the second step. We aim to see that specifically the removal of the GE component should have a negative impact on performance.

### 3.2 VALIDATION WITH SYNTHETIC DATA

In this section we create synthetic networks in which the data clusters are obvious and we test the ability of our pipeline to recover them.

We create a stochastic blockmodel network (SBM) with $|K|$ communities, each containing 50 nodes. The average degree of the nodes in the SBM is equal to 20. Each node has, on average $d_{out}$ connections pointing outside its own cluster and $20 - d_{out}$ connections pointing to inside the cluster. Each

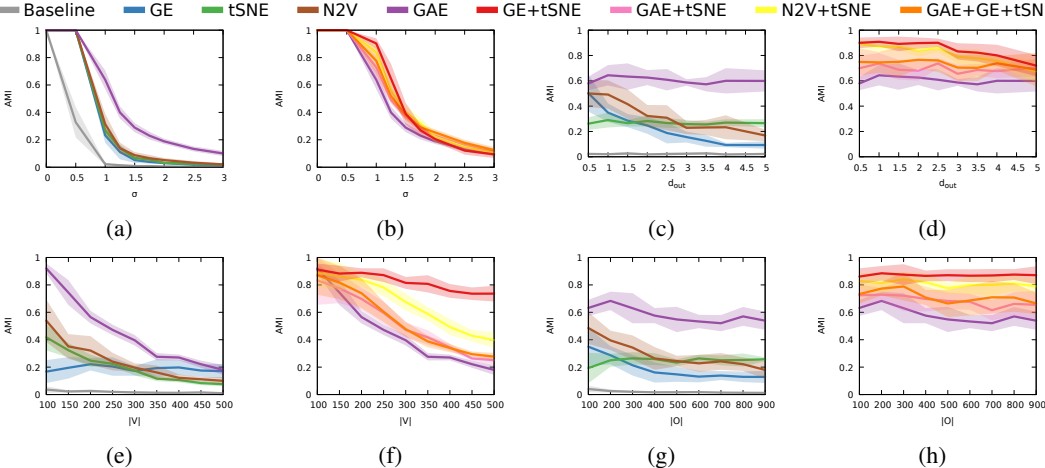

Figure 3: The AMI score (y axis) of the various methods (line color). Average performance as the line, standard deviation as the shaded area. x-axis from left to right increases: (a-b) observation noise $\sigma$; (c-d) network connections noise $d_{out}$; (e-f) node count $|V|$; (g-h) observation number $|O|$.

observation $o$ is a vector of length $|V| = 50|K|$ and corresponds with a $k_o \in K$ community in the graph. The values in $o$ are extracted from two random uniform distributions. The values corresponding to nodes that belongs to $k_o$ comes from a random uniform in the domain $[.5:1[$, while values corresponding to nodes from outside $k_o$ come from the domain $[0:.5[$. Therefore, we expect that $O$ has $|K|$ natural clusters $C$, with each $c \in C$ corresponding to a $k \in K$ – we can therefore evaluate the clustering performance via the Adjusted Mutual Information (AMI) Vinh et al. (2009) between the clusters we obtain and the pre-planted communities of the network. To estimate how resilient to noise the methods are, we apply a gaussian noise to each $o$, coming from a normal distribution with average zero and standard deviation $\sigma$. The higher the $\sigma$ the more noise there is and the performance of the clustering methods should decrease accordingly.

We investigate performance across different values of noise in the observations ($\sigma$), noise in the network structure ($d_{out}$), size of the network ($|V|$), and number of observations ($|O|$). For each experiment we change the focus parameter keeping the others to their default values, which are: $\sigma = 1$, $d_{out} = 2$, $|V| = 200$, $|O| = 300$. We repeat each experiment for 10 independent runs and we report average and standard deviation of the results.

We start by analyzing the effect of noise in the observations ($\sigma$). We start from Figure 3a, showing the results with each method in isolation. First, all methods vastly outperform the baseline: in this network setting, not knowing about the underlying network and assuming that dimensions are unrelated leads to poor performance. The only exception is when $\sigma = 0$: if there is no noise at all, the network information is irrelevant. However, this is a wildly unrealistic scenario.

Second, the method performing by far better than anything else is GAE. As expected, performing embeddings with a deep graph neural network is the current state of the art.

Finally, all other methods (tSNE, GE, and N2V) perform roughly in the same class. N2V has a slight edge, showing how even shallow graph embeddings are well performing. However, tSNE dimensionality reduction is powerful enough to be on par performance with other network-aware methods, even if it does not consider any network information at all.

We now move to the second step (Figure 3b), testing our composite framework. We replicate the GAE performance, to contextualize between the two analysis. For high levels of noise, there is no large difference between the various methods, with the full GAE+GE+tSNE framework ranking first. However, if noise is not strong enough to completely swamp the signal in $O$, then the best performing method is actually the combination of GE with tSNE. This is a genuine synergy between the two methods, because adding the GAE to them lowers performance, and the GAE+tSNE method is strictly lower than GAE+GE+tSNE. In this scenario, the GE component is fundamental to achieve optimal performance, unless high levels of noise make GAE+GE+tSNE the preferred option.

We now analyze what happens when the communities in $G$ become less well defined, i.e. the expected degree of a node pointing outside its community ($d_{out}$) grows. This will make the clusters harder to find. In the first step (Figure 3c), we see that the methods that do not take the network as input (baseline, tSNE) are not affect, as expected. GAE is also resilient to network noise. Both N2V and our GE instead are affected by the weakening of communities.

Moving to the second step (Figure 3d), once again, GE+tSNE is the preferred method: GAE does not give a significant contribution to the full framework (GAE+GE+tSNE) and the GE component is necessary – as GAE+tSNE is inferior to GE+tSNE , just like we observed in the previous test.

What if the underlying network grows? This is normally not a problem when clustering nodes – at least for methods that do not have a resolution limit Fortunato & Barthelemy (2007) – however here this is a problem. The reason is because we are not clustering nodes, but observations that live in a network. The nodes of the network represent the dimensions of the space in which these observation live. As a consequence, a larger network will lead to a harder clustering tasks, because it is harder to cluster high dimensional data than low dimensional one.

Figure 3e shows this principle, as we increase $|V|$. tSNE's performance degrades even if it does not take $G$ into account, because a higher $|V|$ means more dimensions that need to be summarized. All methods degrade their performance: N2V does not get as much information out of its random walks on the overall structure, and GAE has more dimensions to handle in the encoding-decoding layers.

The sole exception is our GE method, which is indifferent to $|V|$ and offers constant performance. While this is not useful for smaller networks, it gets more and more significant as $|V|$ grows. As a result, Figure 3f shows that GE+tSNE is by far the best method, which is even more relevant considering that most real world networks tend to be large. GAE dominates the method when introduced and so the full framework's performance degrades with larger networks.

A final question centers on the effect of the number of observations $|O|$. It is possible that increasing the size of $O$ improves performance, as the methods have more and more data to identify the latent patterns. However, in this case neither Figure 3g nor Figure 3h show an appreciable difference for each method as $|O|$. The ranking of the composite frameworks in the second step is maintained, with GE+tSNE performing best overall – a constant across all tests that we run.

In Appendix B we sum up the tests quantitatively, showing how GE+tSNE is the preferred approach.

### 3.3 VALIDATION WITH REAL WORLD DATA

We use real world data with ground truth to validate the performance of the network embeddings with unknown and noisy data generation processes. We use two case studies using the Trade Atlas and the Little Sis datasets. Since all methods except Baseline and GE have random fluctuations, we repeat the experiments 25 times and we show the distribution of the resulting performances.

### 3.3.1 TRADE ATLAS

The data originates from the United Nations Comtrade datasets. We obtained it through the Harvard dataverse at Harvard University (2019). After the data cleaning procedure discussed in the Appendix, we obtain $G$ as a simple undirected network connecting two countries with the total trade volume in either direction across all traded products. Each vector in $O$ is a product and the values in the vector are the amount exported by each country for this product. To deal with the highly skewed nature of this data, we take the logarithm of export values and we standardize it.

We can use the network embeddings to cluster $O$ using the information in $G$. The objective is to reconstruct the product category, under the assumption that countries specialize their productive activities according to the knowledge they have, which is more easily transferred across products in the same macro category – this is the base of economic complexity theory Hausmann et al. (2014).

From Figure 4a we can see that GAE works extremely poorly, perhaps because the relationship between the node attributes and the graph structure is non-linear and too complex. GE network embeddings, by themselves, are underwhelming, actually performing on par with the baseline. However, using them to provide the embeddings to calculate tSNE provides a significant performance boost to tSNE alone, showing their effectiveness when combined with dimensionality reduction techniques.

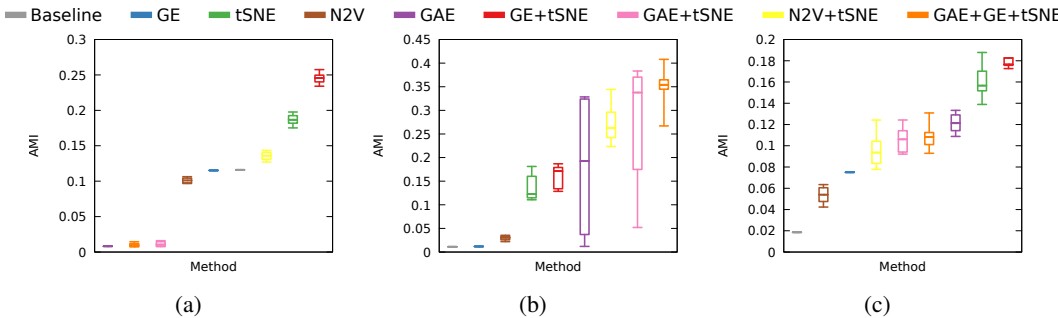

Figure 4: The AMI score (y axis) for the different methods (x axis and color). The boxplots show the 10th, 25th, 75th, and 90th percentile, along with the average performance. We sort the methods left to right in ascending average performance order. (a) Trade Atlas, (b) Little Sis, (c) Tivoli.

### 3.3.2 LITTLE SIS

For this validation, we want to infer a politician's ideology by looking at the social network of their political donors. The data originates from LittleSisLittleSis (2022) a nonprofit organization tracking family, social, and work links between worldwide elite people. We perform several rounds of data cleaning, which are described in the Appendix. We end up with a social network $G$ of political donors with 529 nodes and 871 edges. In $O$, each observation is a member of the current US Congress. The values in their vectors are the amount they received in campaign donations from each of the 529 donors in $G$.

In this case, the ideology is represented by their party affiliation – either Republican or Democrat – which we can use for validation (how well can our cluster reconstruct the US Congress parties?). Figure 4b shows the result. Once more, GE by itself performs on par only with the Euclidean baseline – which is to say, close to zero AMI. Among all the isolated methods, GAE has the best average performance, but it is highly erratic: it can return highly aligned (AMI $> 0.3$) or highly misaligned (AMI $\sim 0$) clusters depending on random fluctuations.

This is where our GE network embeddings can help. Combining GAE with tSNE alone improves the average performance, but retains the erratic behavior. Instead, if we add the GE space to the tSNE embeddings and GAE, we obtain the highest average (and maximum) performance, with a much reduced variance.

### 3.3.3 TIVOLI

For our application, we focus on the task of product recommendations to customers. The data comes from the amusement park Tivoli: $G$ is a network of rides connected by weighted edges counting how many people holding a given pass rode on both (Figure A1). Pass information composes $O$, which is a vector of how many times each pass checked in a given ride – details on data cleaning are in the Appendix. Each pass has a type – regular, school, children, etc – which is our ground truth. The objective is to find clusters of passes aligned with their type. If successful, it means we can infer the type a pass behavior corresponds to, meaning we can create new product bundles and suggest to new customers the best product upgrade they should purchase, given their behavior in the park so far.

Figure 4c shows the result. Again, the best performing method is the GE when combined with tSNE, showing that we can get better recommendations for pass purchases to customers.

### 3.4 EFFICIENCY

Calculating the network-aware embeddings via GE can scale to large networks. There is no need to calculate $L^+$ explicitly. One can estimate the distance between two node attribute vectors by using Laplacian solvers Spielman & Teng (2004); Koutis et al. (2011); Spielman & Teng (2014), which brings the time complexity of the method down to near linear time regime – in number of nodes.

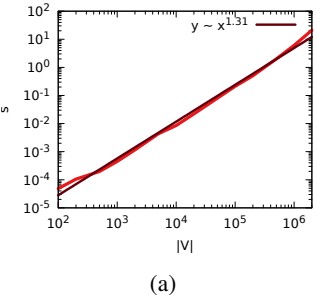 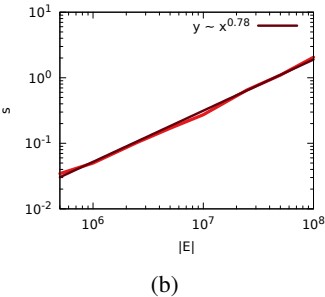

|       |       |
|:-----:|:-----:|
| (a)   | (b)   |

Figure 5: The runtimes (y axis) for benchmark networks of growing sizes (x axis). Actual runtimes in bright red, best fit in dark red.

To show this, we perform two experiments. We specifically use the gaussian elimination Laplacian solver Kyng & Sachdeva (2016), but the difference in runtime with other solvers is negligible. We create a benchmark using stochastic blockmodels as we did for the experiments in Section 3.2. We use a Julia implementation and run the experiments on a Intel Xeon W-11955M CPU at 2.60GHz.

First, we fix the number of clusters and average degree to $4$ and we create larger and larger SBMs in number of nodes – from $100$ to $2,000,000$. Figure 5 shows the result. From Figure 5a, we have confirmation that the runtime grows faster than $|V|$, but decisively less than $|V|^2$. Given the data we have, the best empirical fit of the runtime scaling is $\mathcal{O}(|V|^{1.31})$.

We also fix $|V| = 50,000$ and test the effect on the runtime of having denser and denser network, by increasing $|E|$. From Figure 5b, we see that the runtime is actually sublinear in terms of $|E|$. Given the data we have, the best empirical fit of the runtime scaling is $\mathcal{O}(|E|^{0.78})$.

Note that using the Laplacian solvers is not necessarily the best option. It is advisable to do so only for large networks of, say, $|V| > 10,000$. Below that size, it might be a better idea to actually calculate $L^+$ and cache it. Since $L^+$ is the same for any given $G$, if $G$ is small but there are many attributes for which one wants to calculate their GE distances, then re-using $L^+$ would be faster than using the Laplacian solvers.

## 4  CONCLUSIONS

We introduce a new way to perform data clustering. Specifically, we create the notion of network embeddings: to create embedding of node attributes. In this scenario, the observations are values attached to nodes, and the underlying graph determines the relationships between the dimensions of analysis. This is a new type of unsupervised learning that has hitherto received little attention.

In the paper we show how to calculate network embeddings using effective resistance and the generalized Euclidean distance. We use these embeddings in a pipeline that cleans node attribute data via a graph autoencoder, performs dimensionality reduction using tSNE, and finally detects clusters of node attributes using DBSCAN. Experiments show that the network embeddings, by themselves, are not particularly useful, reaching performances achievable with tSNE without any notion of the underlying graph. However, when combined in the larger pipeline, they lead to significant improvements in performance over the state of the art. These improvements are consistent across various analytic scenarios. Our case studies point at a number of potentially interesting applications of this new data clustering problem and technique.

Potentially, this is the first step in the creation of a new sub-branch of data clustering. Future works include: the refinement of the pipeline, by optimizing each of its components; the exploration of new ways of calculating network embeddings, using other generalized network distances techniques Coscia et al. (2020); and new applications, deepening the exploration of our case studies with domain experts, who can interpret and contextualize our results.

## REPRODUCIBILITY STATEMENT

All code and data necessary to reproduce the main results of the paper are publicly available as supplementary materials for this paper and at [URL redacted for double blind review]. This includes everything needed to reproduce all subfigures from Figures 3, 4, and 5.

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

## A  DATA CLEANING

### A.1  TRADE ATLAS

In this dataset, exporters connect to importers, recording the value of each good (SITC classification) traded each year. We filter out countries and products below a minimum trade volume threshold. We select data from 1993 until 2013, keeping only countries that appear exporting/importing at least one product in each observed year. We project over the time dimension, resulting in an exporter-importer-product tripartite network summing all trade links in the 1993-2013 period, to smooth out fluctuations. Our final dataset connects 160 exporters/importers through the 699 products they traded.

### A.2  LITTLE SIS

To create a meaningful $G$ network, we select a set of valid donors, which will be the nodes of $G$. To be a valid donor, an entity (which can be a person or an organization) must have donated at least 1,000 USD dollars to an elected member of the current (118th) US Congress. This donation must be marked as "current" in the LittleSis data, or it has to be relatively recent – in 2020 or later. This gives us a set of 1,376 potential donors.

We derive the edges of $G$ in a two-step procedure. First, if two donors have a family (father, mother, brother, etc) or social (spouse, friend, etc) link, we connect them. This generates 725 edges. Second, two donors can be connected if they have a significant amount of indirect relationships: sitting on the board for the same companies, being partners in the same ventures, etc. We only consider the edges

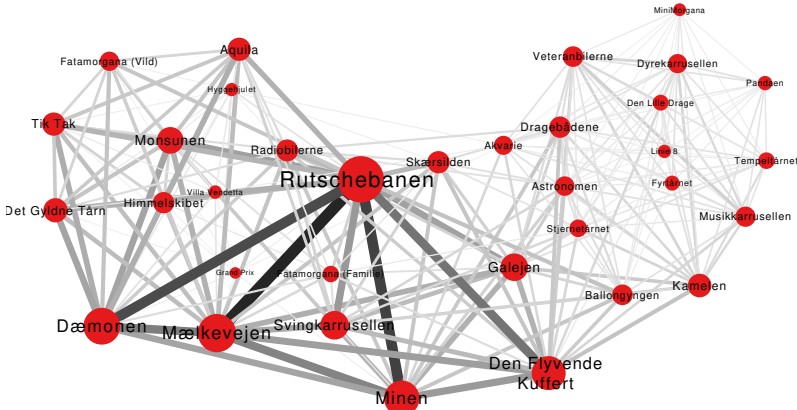

Figure A1: The network from the Tivoli dataset. Node size is proportional to the number of customers taking the ride. Edge size and color is proportional to the number of customers taking both rides (dark = high number, bright = low number).

that have a statistically significant weight, using the noise corrected backboning method Coscia & Neffke (2017) – which is especially designed for exactly this type of data, with discrete edge weights representing counts. This results in 300 additional edges.

Finally, since we can only calculate network embeddings on a connected graph, we select $G$'s giant connected component. This results in a $G$ with 529 nodes (donors) and 871 edges.

### A.3 TIVOLI

In this dataset, the graph $G$ connects rides in the amusement park, with weighted edges counting how many passes checked in on both rides. We have data on 34 rides – which is the number of nodes in the network. The total number of passes from which we generate the network is 629k. We again establish significance using the noise corrected backboning method Coscia & Neffke (2017). After filtering, $G$ contains 244 edges, which implies high density.

From Figure A1 we can see that there is an interesting cluster structure. This is informative, as the cluster on the right is predominantly composed by rides targeted to young children. The edge weights are broadly distributed, so that using the weight information should help in exploiting the topology of the network.

For $O$ we sample around 600 passes. Each $o \in O$ is a vector telling how many times a given pass checked in on a given ride. The label we want to predict is the type of the pass – a tour pass, a premium tour pass, a kids ticket, etc. The sample is made randomly, but representatively of the different pass types, so that the relative popularity of each pass type is respected. For instance, if $x\%$ of the passes are regular tour passes in the data, then $x\%$ of the passes sampled are of the regular type. We need to drop four pass categories that are not popular enough to be included in the sample, leaving us with a total of seven categories to predict.

## B SUMMARY PERFORMANCE

We sum up the validation from Section 3.2 in Table A1, which shows how GE+tSNE is the best approach across all dimensions. GE+tSNE is the preferred method to cluster observations of node attributes on a network. This is true for all observation counts $|O|$, network sizes $|V|$, and network community noise levels $d_{out}$. GE+tSNE is only matched by the state of the art (GAE) for high levels of noise in the observations ($\sigma$). In those cases, GAE could be preferred, although we would argue that the GE+tSNE approach is simpler, as it does not require deep learning.

| Method | $\sigma$ | $d_{out}$ | $|V|$ | $|O|$ |
|---|---|---|---|---|
| Baseline | 0.155 | 0.022 | 0.019 | 0.023 |
| GE | 0.278 | 0.226 | 0.189 | 0.219 |
| tSNE | 0.290 | 0.268 | 0.196 | 0.243 |
| N2V | 0.301 | 0.322 | 0.238 | 0.328 |
| GAE | 0.442 | 0.605 | 0.449 | 0.603 |
| GE+tSNE | **0.514** | **0.853** | **0.823** | **0.871** |
| GAE+GE+tSNE | 0.503 | 0.734 | 0.534 | 0.728 |
| GAE+tSNE | 0.487 | 0.696 | 0.517 | 0.707 |
| N2V+tSNE | **0.511** | 0.814 | 0.661 | 0.811 |

Table A1: The average performance of each method in Figure 3.

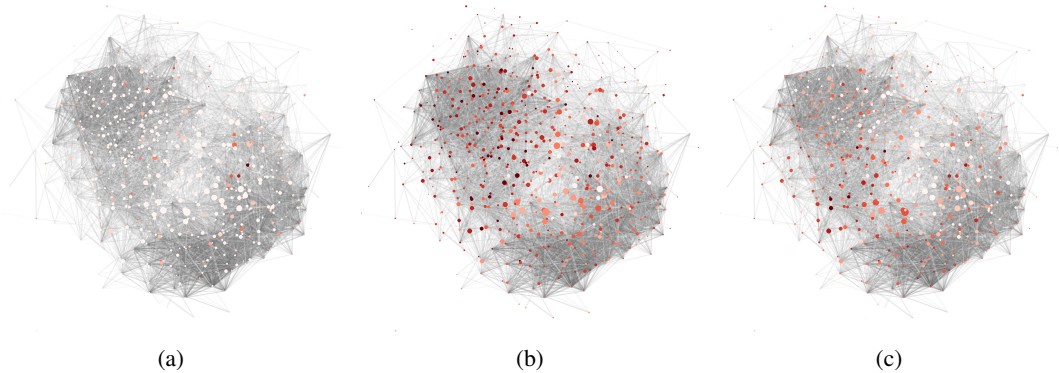

(a)  (b)  (c)

Figure A2: Some of the clusters we found on the Product Space. Nodes are products, edges connect products co-appearing frequently in the same export baskets. Node color is the average export volume for the countries in a given cluster.

## C  ADDITIONAL APPLICATIONS

### C.1  MACROECONOMICS

In Section 3.3 we show, during validation, how network embeddings can be used to reconstruct the SITC classification of products traded in the world's economy. It follows that network embeddings can generate a new product classification, rather than reconstructing the already existing one. In this section, we showcase that network embeddings can also solve the orthogonal problem: to classify countries according to what they export.

To do so, we use as $G$ our reconstruction of the Product Space: a network connecting products if they are co-exported in significant amounts by the same countries Hausmann et al. (2014). Rather than using the original Product Space topology, we calculate our own using the network backboning method, since we need it to focus only on the products and the time period we consider in this paper. In this scenario, each observation in $O$ is the total export basked of a country in the 1993-2013 period. Clusters in $O$ highlight groups of countries with similar export baskets that occupy related areas of the Product Space.

The clusters we find need validation from human experts to gauge their usefulness, since here we do not have a ground truth for automatic validation. We discuss a few examples from Figure A2, showcasing how our results could be used in macroeconomics.

Figure A2a shows cluster #2, which is composed by countries such as United Arab Emirates, Iraq, Kuwait, Saudi Arabia, and so on. The figure shows the extreme concentration in export volume in a handful of products: SITCs 333 (oil) and 341 (gas). We can contrast cluster #2 with cluster #17 in Figure A2b. Cluster #17 is more diversified. It includes economies such as Germany, the UK, and the USA. They occupy most of the Product Space, with diverse top exporter products such as SITCs 541 (pharmaceuticals), 781 (cars), and 776 (electronics).

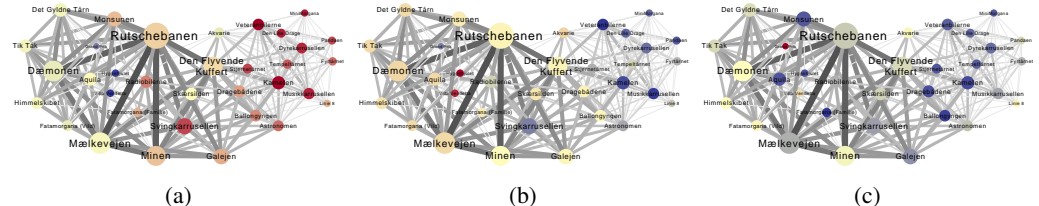

(a)          (b)          (c)

Figure A3: Same legend as Figure A1. Node color is determined by how much the ride is used in a given cluster (red = above average, yellow = average, blue = below average).

Not all diversified countries fit the same mold, as they can focus on different areas of the Product Space. An example is cluster #13 (Figure A2c): it includes other large economies such as China, Malaysia, Thailand, and so on. While diversified, this cluster is particularly strong in specific areas of the Product Space, its top products being 776 (electronics), 752 (computer and parts), and 764 (telecommunication equipment).

Finally, note that clusters tend to be clustered geographically because neighbors tend to specialize in the same products and have knowledge spillovers with their neighbors Bahar et al. (2014). However, not all clusters follow this logic if there are strong similarities that transcend geography. An example is cluster #19 (not depicted) which groups Israel and Singapore, which combines the strengths of cluster #17 (being strong on pharmaceuticals) and cluster #13 (being strong in telecommunication equipment).

## C.2 POLITICAL IDEOLOGY

In Section 3.3 we validate network embeddings by calculating the AMI between the clusters we can extract with them and party memberships. Using the full GAE+GE+tSNE framework we can obtain relatively high AGI values ($> 0.4$), which means there are still several representatives that are misaligned. Here we zoom into these "errors" to see whether they actually carry valuable information that is not captured by party membership.

While validating the entirety of the clusters requires a large expertise in political science, we can use two cases to exemplify the values of the clustering "mistakes".

The first is that of Henry Cuellar, who is the lone Democrat in an otherwise pure Republican cluster. This should not be a surprise, because Cuellar is considered one of the most conservative Democrats and he is part of the Blue Dog caucus, representing the center-right in the Democratic Party whose members are mostly elected in Republican-leaning districts. In fact, Cuellar is a representative from Texas, a traditionally Republican state. With a NOMINATE score Poole & Rosenthal (2000) of $-0.228$, Cuellar is considered more conservative than 92% of Democrats currently in Congress[2].

The converse case is that of Glenn Thompson, a rare Republican in an overwhelmingly Democrat cluster. Thompson has a NOMINATE score of $0.322$, making him more liberal than 88% of Republicans[3]. Just like Cuellar, Thompson is part of one of the most moderate caucuses of his party: the Republican Governance Group.

## C.3 CUSTOMER SEGMENTATION

Figure A3 shows some of the clusters we can extract from the Tivoli data with our method. Figure A3a is cluster #1, which is the most populated. It comprises the "standard" visit to Tivoli, which happens to put together regular passes with "Mini" passes geared towards younger people. Clusters #2 and #3 (Figures A3b and A3b) shows a more specialized interest to several rides outside the cluster on the right. One potential insight, is that currently customers are using regular passes and "Mini" passes in similar ways that are lumped together in the same cluster, which might point at the need to differentiate these offers more.

---

[2]https://voteview.com/person/20533/henry-cuellar
[3]https://voteview.com/person/20946/glenn-thompson

