# OpenReview forum: "Unsupervised Learning via Network-Aware Embeddings"
_ICLR.cc/2024/Conference — Submitted to ICLR 2024_

### Official Review · Reviewer_b2DR · 2023-10-27

**Soundness:** 2 fair
**Presentation:** 3 good
**Contribution:** 2 fair
**Rating:** 3
**Confidence:** 4

**Summary:**

This paper proposes an effective clustering approach for graphs with node attributes. To clean noise, the proposed approach uses Graph Autoencoder. It then uses t-SNE to reduce the dimensions of attributes. After these processes, it computes clusters by using DBSCAN. The paper conducted experiments to show the effectiveness of the proposed approach.

**Strengths:**

- The clustering for attributed graphs is an important research problem in the field.
- This paper is well structured and easy to follow.
- The related works are well presented in the paper.

**Weaknesses:**

- The proposed approach is a combination of existing methods.
- The technical depth of the proposed approach is not so deep.
- Simple model to be easily implemented.

**Questions:**

I like the paper's motivation; it is quite a fundamental research problem to improve the clustering accuracy of attributed graphs. However, regarding technical depth, the proposed approach is a combination of the previous approach; it is technically somewhat shallow. This paper should show theoretical analyses of the proposed approach. Does the proposed approach have any theoretical properties on the clustering result?

The computational costs of t-SNE and DBSCAN are quadratic to the number of data points. What is the theoretical computational cost of the proposed approach? In addition, from the results shown in Section 3.4, the running time of the proposed approach is less than |V|^2. Please tell me why this happens.

Although several GNN-based clustering approaches are listed in Section 1, none of them are compared in the experiment. Is the proposed approach more effective than the other approaches?

The experiments use only two real-world datasets (the Trade Atlas and the Little Sis datasets). Is the proposed approach effective in other real-world datasets?

Even though the proposed approach is a combination of the existing approaches, such as autoencoder, t-SNE, and DBSCAN, their parameter settings are shown in the paper. Please describe the detailed parameter settings.

---

> ### Author Response · Authors · 2023-11-13
>
> We are glad to see the reviewer understanding and praising the motivation and problem definition.
>
> We do agree that the technical contribution is shallow. In our intention, the technical contribution is limited to show that a dedicated approach to solving this innovative problem can outperform adapting other embedding methods. In a sense, the simplicity is a feature: it makes the approach easier to understand and shows that there are large gains to be made once we establish that this is a path worth following. The paper aims to establish that this is a path worth following, optimizing the performance is worthwhile to do only once we have established this.
>
> Q: Does the proposed approach have any theoretical properties on the clustering result?
>
> A: We have not come around proving theoretical properties just yet. Some can be easily included, for instance we can prove that GE is a proper metric in the G space, a guarantee that other methods do not have.
>
> Q: What is the theoretical computational cost of the proposed approach?
>
> A: The theoretical computational cost of the proposed approach is |O|^2 + |V|^x, with x dependent on the exact Laplacian solver used, but normally x ~ 1.3, decisively below 2. The |O|^2 term comes from the correct observation from the reviewer: tSNE and DBSCAN are quadratic and they are applied to the node attributes O. In most real world scenarios, |O| <<< |V|, we normally have much fewer node attributes than nodes. If that's the case, |O|^2 disappears and the running time is dominated by calculating the GE distances, whose complexity is below |V|^2, as the figure shows.
>
> Q: Is the proposed approach more effective than the other [GNN] approaches?
>
> A: We fully expect it to be so. That is because the referred GNN approaches mentioned in our discussion all share the same issue: they were not defined to originally solve the problem we propose. We are arguing that the adaptation of any of those methods will be outperformed by applying GE, since it is designed to tackle this problem explicitly. We have done some experiments resulting in lower performance with other approaches such as Graph Convolutional Networks and Graph Attention Networks. We have not included them in the paper not to have too many comparisons, which would make the figures less clear. We can consider replacing node2vec with one of these methods, or to expand the appendix with these results.
>
> Q: Is the proposed approach effective in other real-world datasets?
>
> A: There is also a third dataset, Tivoli, which shows the same patterns as the previous two real world dataset. So the answer to this question is yes. We can make an effort to find more datasets, if this is considered crucial.
>
> Q: Please describe the detailed parameter settings.
>
> A: We can definitely expand on this on the paper. We overlooked such detail, because we largely use default values. The exception being setting min_samples for DBSCAN to 2, and doing a grid search to find the best performing eps parameter value. We also discuss the setting of the two key parameters for node2vec in Sec 3.1. In light of our answer to reviewer x6pK, we will also discuss tSNE's parameter perplexity better, since we will make a study of it.

---

### Official Review · Reviewer_x6pK · 2023-10-30

**Soundness:** 3 good
**Presentation:** 3 good
**Contribution:** 2 fair
**Rating:** 5
**Confidence:** 3

**Summary:**

This paper proposes a method to cluster a set of network node attributes generated from the same network instead of clustering network nodes.
They measure the network-aware distance between node attribute vectors by the generalized Euclidean distance using the pseudoinverse of graph Laplacian as proposed by Coscia (2020).
Then they perform dimensionality reduction by using tSNE and perform clustering by using DBSCAN. They also propose to apply GAE to clean noise.
Experiments demonstrate that the combination of the generalized Euclidean distance with tSNE improves the quality of clustering obtained from DBSCAN.

**Strengths:**

- The authors tackle the rather unique problem of clustering node attribute vectors rather than clustering nodes of a network.
- The proposed method is simple and efficient. If the noise is not strong enough then GAE is not needed.

**Weaknesses:**

- The theoretical ground of the good clustering performance of GE+tSNE is not clear. It may be to much to say GE+tSNE is a good choice for clustering while still you can say that GE+tSNE is a good choice to visualize graph attribute vectors.
- The clustering with DBSCAN is performed in two dimensional space. If clustering in such a low dimensional space works, then the data may be too simple and too easy to solve. In fact, the true number of categories, which is the target number of clusters, in LittleSis data is small -- just two: either Republican or Democrat. The true number of categories in Trade Atlas and Tivoli are not reported but may be small as well.
The findings here might not generalize well to other more complicated datasets.

**Questions:**

The tSNE has a tunable parameter called perplexity. I would like to know whether the results are robust with different perplexity parameter values.

It is unclear tSNE is the best choice. The use of tSNE into low, 2 or 3, dimensions may be good for visualization but not for clustering. For clustering,  the use of UMAP into relatively higher dimensions can be a better choice.

**Details Of Ethics Concerns:**

I have no concerns.

---

> ### Author Response · Authors · 2023-11-13
>
> Thank you for the positive review.
>
> About the comment on 2D spaces to be easy to cluster: that is certainly true. However, ending up with an easy 2D clustering problem is exactly the power of the pipeline. Originally, O -- the data we want to cluster -- has n dimensions, with n being the number of nodes in the graph G. n can be very large if we are looking at a social graph -- up to millions or tens of millions of nodes. With our pipeline, we can make sense of these millions of dimensions, reducing them to a low and manageable number. To us, that is a strength of the method.
>
> About the number of categories of Trade Atlas and Tivoli, we can add this information to the paper. For Trade Atlas, the "product categories" we refer to is the first digit of the hierarchical SITC codes -- there are ten product categories in total. For Tivoli, there are seven categories.
>
> As for the direct questions:
>
> - We can definitely check the impact of perplexity on the results. We will update the review once we manage to run such test, which we can include in the appendix.
>
> - We agree UMAP could be a better choice than tSNE. We can use UMAP without issues -- we already have the code to do so. If the reviewer feels strongly about it, we'll do so. However, we point at the last paragraph of Sec 2.4.3 as to why we didn't do it in the first place: we aren't at this point trying to maximize every piece of our framework. Our focus is on defining this innovative problem definition and showing that a dedicated way to tackle it -- using GE -- outperforms adapting other embedding methods that were not developed with this problem in mind. Plugging in UMAP rather than tSNE will likely produce benefits across the board for all methods, whether they use GE or not, which is not a current focus of the paper.

---

> > ### Comment · Reviewer_x6pK · 2023-11-22
> >
> > Thank you for the detailed explanations.
> > I still think more theoretical analyses are needed to verify the arguments.

---

### Official Review · Reviewer_L2DL · 2023-11-01

**Soundness:** 2 fair
**Presentation:** 2 fair
**Contribution:** 3 good
**Rating:** 5
**Confidence:** 4

**Summary:**

The authors introduce an attributed-graph clustering scenario where it is of interest to cluster the attributes of the graph rather than the nodes. They propose a pipeline that involves a graph-based attribute distance metric, a graph auto-encoder, dimensionality reduction via TSNE, and a final clustering step. They devise a synthetic benchmark framework to stress-test their approach vs baselines and ablations. They also evaluate the pipeline & alternatives on three real-world datasets from a variety of domains.

**Strengths:**

(1) The paper's main strength is the novelty of the problem setting. I am not aware of any other approach to clustering that attempts to use an explicit graph over the dimensions. This is an interesting setting and worth exploring.

(2) The paper's simulation study and real data studies are (each) thorough, well-described, and appropriate for evaluation.

**Weaknesses:**

I will elaborate on these high-level weaknesses (with evidence) in my questions to the authors.

(1) The paper lacks a clear understanding of how each of the individual modules are used. In some places, there is reason to question the soundness of the modules that are proposed.

(2) The proposed pipeline has unclear efficacy on the real-world datasets. It is not clear to practitioners whether the proposed approach should be used in general, or whether the baselines would suffice.

(3) The authors discuss a scalable estimation of the inverse Laplacian, but they do not check the performance of the approximation in their results.

**Questions:**

It is very unclear how the graph auto-encoder (GAE) is applied in the pipeline. A graph autoencoder for attributed graphs performs node feature convolutions over the graph to arrive at a $n \times d_a$ representation of the graph, where $n$ is the number of nodes and $d_a$ is the size of the inner hidden dimension. It is possible to reconstruct an $n\times n$ low-rank estimated graph from this latent representation. We must consider this in light of the authors' statements:
 * Q1: The authors describe the GAE baseline as follows: "GAE: clusters $O$, after passing it through our graph autoencoder, described in Section 2.4". It is not at all clear how one can "cluster $O$" from the output of the GAE. To cluster $O$, you need to cluster the columns of some matrix with column dimension $d(O)$. Since the output of a GAE is either $n\times d_a$ or $n\times n$, it is not at all clear how one can "cluster $O$" after "passing it through" a GAE. Can the authors elaborate?
 * Q2: In S2.4.2, the authors write "We could apply tSNE directly to the GAE output." As above, this lacks a formal description, and it is ambiguous how this could actually be achieved. Again, we would need to apply tSNE to some $k \times d(O)$ matrix, which would result in a tSNE-estimated $2 \times d(O)$ matrix. How does the GAE output allow for this?
 * Q3: Figure 2 implies that somehow the GAE output is used in the distance metric computation. However, this is never described (even vaguely). The formula for the distance metric (given in Section 2.3) includes only the original attribute data $O$ and the inverse laplacian $L^{+}$, which is derived from the original graph $G$. How does the GAE output feed into this?
 * Q4: The authors write "We could apply tSNE directly to the GAE output. However that would mean that tSNE is seeking for the best representation of the data in an Euclidean space, which is not appropriate because we know the dimensions of our observations are related to each other in $G$". The plausibility of this reasoning is hard to evaluate because we don't know how the GAE is being applied. It is quite possible that (depending on how the GAE fed into the tSNE module), tSNE is actually *not* being applied in a "Euclidean" space, because the output of the GAE directly convolves the node features with the graph structure. Can the authors please clarify this statement, as well as the application of GAE itself?

Q5: The only justification the authors give for their distance metric, which is arguably a core part (if not *the* core part) of the contribution, is very weak:

"Previous work shows that this formula gives a good notion of distance between $o_1$ and $o_2$ on a network Coscia (2020). For instance, it can recover the infection and healing parameters in a Susceptible-Infected-Recovered (SIR) model by comparing two temporal snapshots of an epidemic."

To the best of my understanding, there is no immediate connection between a metric's ability to estimate SIR parameters and a metric's ability to cluster data well. Can the authors please elaborate their justification for this choice?

Q6: While the approximation to $L^{+}$ seems efficient (see Section 3.4), the authors did not explore how the approximation affects the performance of the algorithm. Could the authors please replicate the simulations and real data analyses using the approximation to $L^{+}$ rather than the exact solution? This would be a welcome addition to the appendix.

Q7: From a methodological perspective, the conclusions that a practioner can draw from the empirical study is very limited. This is because the baselines of the proposed pipeline do almost as well as the pipeline on the real-world data. In particular, for each dataset, two out of the top-3 scoring approaches do not use the proposed graph distance at all. Again on each dataset, ablations that *do* use the graph distance have at-most middle-ranks, and one of the scores for the full pipeline (GAE+GE+tSNE) is near-random performance (see Fig 4(a)). Can the authors further justify the use of their pipeline in light of these results?

If the authors can answer these questions satisfactorily and provide a substantial revision, I might be able to raise my score.

---

> ### Author Response · Authors · 2023-11-13
>
> We thank the reviewers for these in-depth questions and we are glad to see they see the innovation of the method.
>
> Q1: Can the authors elaborate [how GAE works]?
>
> A1: Saying that "GAE clusters" something is sloppy on our part and we'll fix it. What we mean here is that we build a pipeline which applies GAE to O, obtaining a transformed GAE(O), and then uses DBSCAN on GAE(O) to find the clusters. GAE is applied to each o_x in O. Each o_x is a vector of length n, which get first encoded and then decoded by GAE to obtain a new GAE(o_x), again of length n. GAE(O) is the collection of such vectors, which is a matrix of dimension n x d(O), as requested -- because we have d(O) o_x (and GAE(o_x)) vectors.
>
> Q2: How does the GAE output allow for [having a k x d(O) matrix]?
>
> A2: Hopefully, the answer above helps with this question as well. Since GAE outputs GAE(O), an n x d(O) matrix, then this can be easily fed into tSNE to reduce its dimensions to 2 x d(O).
>
> Q3: How does the GAE output feed into [the distance calculation]?
>
> A3: This depends on the actual pipeline used. Fig2 shows the full pipeline. Here we first apply GAE to transform O into GAE(O). Then, to estimate the distance between two GAE(o_1) GAE(o_2) vectors we use the formula the reviewer mentioned. Since O and GAE(O) have the exact same shape, they can be used interchangeably. In Sec 2.3 we choose to use O not to have to introduce too many details that might get in the way of understanding what the measure is doing. We can of course change this if the reviewer finds it appropriate.
>
> Q4: Can the authors please clarify this statement [about applying tSNE to the GAE output]?
>
> A4: Hopefully, our A1 helps with this question as well. We also admit that it is true that GAE convolves O and G. So we should point this out in the text. What we still argue is that applying our GE function to GAE's output is, on average, beneficial, as Table A1 shows (compare GAE+GE+tSNE with GAE+tSNE rows).
>
> Q5: Can the authors please elaborate their justification for this choice [of distance function]?
>
> A5: Our main justification for using this distance function is that it provides a strong analogue to the Euclidean distance on a graph. For instance, the paper we cite shows that, in the simple and intuitive toy example of having a chain graph with two vectors o_1 and o_2 occupying the endpoints, the distance measure is the square root of the chain length. The chain length is the number of steps one would need to take on G to go from o_1 and o_2. GE is a proper metric that can be used to replace the Euclidean distance when running tSNE and/or DBSCAN.
>
> Q6: Could the authors please replicate the simulations and real data analyses using the approximation rather than the exact solution?
>
> A6: We can certainly do that. There is also a recently-appeared note on arXiV (https://arxiv.org/abs/2310.11222) showing that the approximations are in the order of 10^-9 (Sec 5.1, Tab1), which we consider negligible.
>
> Q7: Can the authors further justify the use of their pipeline in light of these results?
>
> A7: Our justification is that the application of GE, our distance function, is always beneficial. The top scoring method for all real world datasets always uses GE. What makes the results a bit more confusing is that GAE is not always beneficial and could sometimes be harmful. We think this is due to the amount of noise in the data (as Fig3b shows: adding GAE to GE+tSNE -- orange line -- is only beneficial for high values of sigma -- the noise level). So the justification would be to use GAE+GE+tSNE only if the levels of noise are high, and GE+tSNE when they are not. Using this guideline, one would always achieve top performance in the real world cases we consider in the paper.

---

> > ### Comment · Reviewer_L2DL · 2023-11-23
> > **Reply**
> >
> > Thanks to the authors for their time in responding.
> >
> > Re Q1-Q4, I am a little more clarified, but not completely. I am not familiar with GAEs being used to reconstruct a feature matrix rather than a graph. If a GAE is actually being used in this way, the matrix equations should be expressed in the paper (or at least in the Appendix), as IMO this is not the usual way that GAEs are applied. As it stands, this is not in the current revision, and even the vague "GAE clusters" is still there, so it is hard to assess the future improvement.
> >
> > Re Q5, the description in A5 belongs in the paper. I would like to see this in a revision before raising my score.
> >
> > Re Q6, those approximation results are promising. It would be useful to mention those in the paper.
> >
> > Re Q7, it is still hard for a practitioner to know if the "levels of noise are high". I also do not think the results support the claim "using GE is always beneficial": the GE-only method is a low scorer on all datasets.
> >
> > The suggested edits seem promising, so I will raise my score by 1. However, I cannot raise it further (nor champion for acceptance) unless I can see the edits.

---

### Official Review · Reviewer_uUyU · 2023-11-06

**Soundness:** 1 poor
**Presentation:** 1 poor
**Contribution:** 2 fair
**Rating:** 3
**Confidence:** 5

**Summary:**

This paper proposes that data clustering be done in a manner that combines knowledge of a network connecting nodes and a set of observations O assigning a value to each node. Other than that, the problem definition given and the aim of this paper is unclear.

**Strengths:**

The paper appears to propose a novel way to generate clusters through embeddings.

**Weaknesses:**

The main idea in itself is hardly novel, as embedding-based clustering has been around for a long time. In the experimental study, the paper compares to preliminary embedding methods like node2vec, which have been superseded by other more recent works in the literature. Thus it is hard to judge the success of the method in experimental terms.

More importantly, it is hard to judge this paper because the problem definition is written in a manner that is hard to make sense of. It is claimed that the proposed method clusters node attributes rather than nodes. It is hard to understand what that means, since nodes are represented by their attributes and the method claims to take the network connections in consideration as well.

The problem definition says that we look for a function f : G × O → P returning the partition P such that arg minO δ = f(G, O), δ being the function calculating the distance between pairs of observations on G. Nothing is clear in this definition. For example:

- What is mean to apply a function f on the whole of G and O?
- Where is the pair of observations on which the function δ is applied?
- What exactly is the stated objective expressed as arg minO δ = f(G, O)? Are we looking for a set O or a member of O?
- What should the entity we are looking for minimize?

The explanation offered in words is a bit better than the mathematical statement, yet still unclear: "we want to find the partition P of O such that the graph distance δ over G of observations within the same group in P is minimized – excluding trivial solutions that put each observation in a singleton cluster. This definition hinges on δ: the ability to calculate the graph distances between two o1, o2 ∈ O." The following questions emerge:

- What is the definition of a partition?
- What is the newly named graph distance δ?
- Which exact distance, out of all pairs in a group, would be minimized?
- How are trivial solutions to be excluded?

Given this incompleteness, the paper is unready for publication.

**Questions:**

See weaknesses.

---

> ### Author Response · Authors · 2023-11-13
>
> We agree with the reviewer that the mathematical formulation of the problem could be done better. Part of the confusion might stem from the fact that the problem definition is tricky, as it is very different from what it is normally done with embeddings on graphs. We would encourage to read reviewer L2DL first strength point, since they understood clearly the innovative aspect of it.
>
> We now address the second set of questions, since we think they are the most important to understand the method, and then we can try to improve the mathematical notation accordingly.
>
> Q: What is the definition of a partition?
> A: A partition for us is a grouping of node attributes o in O. Let us suppose we have four node attributes in O: o_1, o_2, o_3, o_4. A valid partition could be one that says that o_1 and o_2 belong to one cluster, and o_3 and o_4 belong to another.
>
> Q: What is the newly named graph distance δ?
> A: In that definition, δ stands as any distance function that takes as input two node attributes (o_1 and o_2) and a graph G. It will return a scalar which represent the distance between o_1 and o_2, over the edges of G. One can think of δ as an Euclidean distance, where the spatial dimensions are related to each other, with G providing these relationships. Movements along dimensions connected in G contribute less to the distance than movements along dimensions not connected in G.
>
> Q: Which exact distance, out of all pairs in a group, would be minimized?
> A: This would be up to the chosen clustering algorithm to decide -- we could have single, average, or complete linkage, or other options. It is a practical question that we feel is not necessary to address in the problem definition itself, because we are describing the problem in general terms (although we are very much open to modify this and to focus on a specific quality function, if the reviewer thinks this is important).
>
> Q: How are trivial solutions to be excluded?
> A: Same as above, we want to introduce the problem in the most general way possible -- akin to define what clustering is, rather than focusing on a specific sub-problem. There could be many ways to exclude trivial solutions. Again, we can modify this part to be more specific, if the reviewer thinks this would strengthen the paper.

---

### Official Review · Reviewer_9U5s · 2023-11-09

**Soundness:** 2 fair
**Presentation:** 1 poor
**Contribution:** 1 poor
**Rating:** 1
**Confidence:** 4

**Summary:**

This paper proposes a node attribute clustering method by considering the graph structure to measure the distance between node attributes. The authors claim that the contribution of this paper is that they are the first to cluster node attributes by considering graph structure.

**Strengths:**

This paper proposes a pipeline to reduce the dimensions of node features.

**Weaknesses:**

1. The biggest problem with this paper is that the model innovation is not enough. In short, the authors propose a distance metric that considers graph structure based on the scheme of Coscia et al. (2020) and apply it to t-SNE to reduce feature dimensions, thereby achieving the goal of clustering node attributes. From my perspective, this does not meet the standard of an academic paper that will be presented at a prestigious conference.
2. The expression in this paper is not concise enough. It is not until the fourth paragraph of the introduction that the main objective of this paper is revealed. The main contribution of the method is the generalized Euclidean proposed in Section 2.3, and many other redundant contents need to be deleted.
3. The expression in this paper is not clear enough. For example, why clean noise, and why can AE clean noise? Why don't other models consider this, and what is the motivation for choosing AE? These are not explained clearly.
4. Even the title is not clear enough. Unsupervised learning is too broad and cannot represent node attribute clustering.
5. The authors should have a section introducing related work.

**Questions:**

Proposed in Weaknesses

---

> ### Author Response · Authors · 2023-11-13
>
> Thank you for your comments. From our understanding, most of the weight that leads the reviewer to give a strong rejection is on point 1. We see points 2 to 5 as easily fixable with simple text editing (we can lift up the main objective description, explain better the role of AE, edit the title, and separate the part of the introduction where we discuss related works into its own section).
>
> About point 1, our perspective is that we agree that the technical contribution by itself is not strong. However, the real contribution of the paper, to us, lays into an innovative problem definition (clustering node attributes using the topology of the network to define the relationships between the dimension of analysis) and the technical contribution represents the only approach specifically dedicated to solve this problem.
>
> We would invite to consider reviewer L2DL first strength point, which acknowledge that this is an innovative and worth exploring problem space.

---

> > ### Comment · Reviewer_9U5s · 2023-11-22
> >
> > Thank you for your response. I still believe that having only the problem definition is insufficient. Also, I have not seen any marked revisions in the manuscript. I will maintain my score.

---

### Meta-Review · Area_Chair_EVy8 · 2023-12-05

**Metareview:**

The authors propose a new method for clustering attributed graph. From a high level perspective the method first a graph-based attribute distance metric followed by a graph auto-encoder and dimensionality reduction via TSNE, and a final clustering step.

The paper contains some interesting idea but it would need major rewriting before being accepted. In particular, the reviewers suggested few weaknesses that should be addressed before re-submission:
- the novelty of the paper is limited
- the problem formulation is not formal enough
- the proposed pipeline has unclear efficacy on the real-world datasets

**Justification For Why Not Higher Score:**

The paper is not yet ready for publications, several weaknesses are highlighted in the reviews

**Justification For Why Not Lower Score:**

N/A

---

### Decision · Program_Chairs · 2024-01-16

Reject